# High-Level Drug-Resistant Mutations among HIV-1 Subtype A6 and CRF02_AG in Kazakhstan

**DOI:** 10.3390/v15071407

**Published:** 2023-06-21

**Authors:** Ainur Sanaubarova, Emma Pujol-Hodge, Natalya Dzissyuk, Philippe Lemey, Sten H. Vermund, Andrew J. Leigh Brown, Syed Ali

**Affiliations:** 1Department of Biomedical Sciences, Nazarbayev School of Medicine, Nazarbayev University, Astana 010000, Kazakhstan; ainur.sanaubarova@nu.edu.kz; 2School of Biological Sciences, University of Edinburgh, Edinburgh EH9 3FL, UK; emma.pujol-hodge@ed.ac.uk (E.P.-H.); a.leigh-brown@ed.ac.uk (A.J.L.B.); 3Kazakh Scientific Center of Dermatology and Infectious Diseases, Almaty 010000, Kazakhstan; dzin72@list.ru; 4Department of Microbiology and Immunology, Rega Institute, Katholieke Universiteit Leuven, 3000 Leuven, Belgium; philippe.lemey@kuleuven.be; 5Department of Epidemiology of Microbial Diseases, Yale School of Public Health, New Haven, CT 06510, USA; sten.vermund@yale.edu

**Keywords:** HIV, antiretroviral drug resistance, DRMs, molecular epidemiology, phylogenetics, A6, CRF02_AG, Kazakhstan, Central Asia

## Abstract

HIV incidence in Kazakhstan increased by 73% between 2010 and 2020, with an estimated 35,000 people living with HIV (PLHIV) in 2020. The development of antiretroviral drug resistance is a major threat to effective antiretroviral therapy (ART), yet studies on the prevalence of drug resistance in Kazakhstan are sparse. In this study on the molecular epidemiology of HIV in Kazakhstan, we analyzed 968 partial HIV-1 *pol* sequences that were collected between 2017 and 2020 from PLHIV across all regions of Kazakhstan, covering almost 3% of PLHIV in 2020. Sequences predominantly represented subtypes A6 (57%) and CRF02_AG (41%), with 32% of sequences exhibiting high-level drug resistance. We further identified distinct drug-resistant mutations (DRMs) in the two subtypes: subtype A6 showed a propensity for DRMs A62V, G190S, K101E, and D67N, while CRF02_AG showed a propensity for K103N and V179E. Codon usage analysis revealed that different mutational pathways for the two subtypes may explain the difference in G190S and V179E frequencies. Phylogenetic analysis highlighted differences in the timing and geographic spread of both subtypes within the country, with A62V-harboring subtype A6 sequences clustering on the phylogeny, indicative of sustained transmission of the mutation. Our findings suggest an HIV epidemic characterized by high levels of drug resistance and differential DRM frequencies between subtypes. This emphasizes the importance of drug resistance monitoring within Kazakhstan, together with DRM and subtype screening at diagnosis, to tailor drug regimens and provide effective, virally suppressive ART.

## 1. Introduction

While an estimated 1.5 million people acquired HIV infection in 2021, the annual number of new HIV infections globally has fallen by 32% since 2010 [1]. The situation in Eastern Europe and Central Asia, however, has been the opposite of the overall global situation, with a 48% increase in HIV infections and the number of AIDS-related deaths being 32% higher than in 2010. In fact, the region of Eastern Europe and Central Asia has the fastest-growing HIV epidemic in the world [1]. Notably, the Republic of Kazakhstan, one of the largest countries in Central Asia by population and area [2], experienced a 73% increase in HIV incidence between 2010 and 2020, with an estimated prevalence of 35,000 people living with HIV (PLHIV) in the country in 2020 [3].

In Kazakhstan, HIV prevention and treatment service coverage remains insufficient. In 2020, 78% of PLHIV were aware of their status, 73% were undergoing treatment (57% of all estimated PLHIV), and 84% of the people undergoing treatment were virally suppressed (48% of all PLHIV). Each of these percentages was far behind the global target of 95%-95%-95% by 2025 [1,3]. Many socioeconomic factors contribute to the alarming state of the HIV epidemic in Kazakhstan, including unsafe injection practices among people who inject drugs (PWID); the limited availability of opioid agonist therapy; and the criminalization of sex work, drug possession, HIV transmission, HIV exposure, and the non-disclosure of HIV infection. Despite decriminalized same-sex sexual relations, the stigma and discrimination against gay men and men who have sex with men (MSM) remain common [1].

The effective administration of antiretroviral therapy (ART) to ensure viral suppression is a key pillar in controlling the HIV epidemic by reducing the infectiousness of PLHIV. The use of specific antiretroviral drugs (ARVs) administered within ART regimens varies. World Health Organization (WHO) guidelines in 2021 recommended 3TC/FTC + TDF + DTG (lamivudine–emtricitabine + tenofovir + dolutegravir) as the preferred first-line regimen for adults and adolescents and 3TC + TDF + EFV (3TC + TDF + efavirenz) as the preferred alternative first-line regimen [4]. In Kazakhstan, current ART guidelines follow the WHO recommendations, though there have been guideline changes in recent years (Table 1) [5,6,7,8].

A major barrier to effective ART is the evolution of HIV drug resistance, which is likely to increase with the continued efforts to scale up ART [9,10]. The development of drug-resistant HIV depends on multiple factors, including the duration of viral replication in the presence of subtherapeutic levels of ARVs, the effect of drug-resistant mutations (DRMs) on drug susceptibility and viral replication, and the ease of acquisition of a particular DRM [11]. HIV drug resistance can be acquired when DRMs emerge in individuals receiving non-suppressive ART or when transmitted by individuals who were infected with drug-resistant HIV [9,12]. In Kazakhstan, no national surveys of HIV drug resistance have been carried out since the time of adoption of newer WHO-recommended guidelines. Earlier studies reported a low prevalence (3%) of HIV drug resistance in therapy-naïve PLHIV who were studied between 2009 and 2013 [13] and a high prevalence (32%) of DRMs in PLHIV receiving ART who were studied between 2017 and 2019 [7].

Despite the steadily increasing incidence of HIV in Central Asia, studies on the molecular epidemiology and phylogenetics of HIV infection in the region remain scarce [14,15,16,17]. In this study, we analyzed sequences from PLHIV across all regions of Kazakhstan, covering almost 3% of the HIV-positive population in the country as of 2020. We quantified levels of HIV-1 drug resistance, identified distinct frequencies of DRMs among viral subtypes, and carried out phylogenetic analyses to elucidate the evolutionary history of the HIV-1 epidemic in Kazakhstan, including the appearance and spread of distinct DRMs.

## 2. Materials and Methods

### 2.1. Ethical Approval, Study Design, and Population

Ethical approval for this study was obtained from the Institutional Research Ethics Committee, Nazarbayev University, Kazakhstan. Data analysis was approved by the ethics committee of the School of Biological Sciences, University of Edinburgh.

Between 2017 and 2020, 1004 participants registered with the Kazakh Scientific Centre of Dermatology and Infectious Diseases, Almaty, Kazakhstan; signed informed consent forms; and consented to blood sample collections for this study. Participants enrolled in the study from all 15 regions of Kazakhstan. Analyses on the CD4+ T-lymphocyte count were performed using the BD FACS Count Reagent Kit™ (BD Biosciences, San Jose, CA, USA), and viral load analyses were performed using the AmpliSens^®^ HIV-Monitor-FRT Kit (InterLabService, Moscow, Russia). A questionnaire was administered to participants to collect information regarding medical history, HIV risk behaviors, and existing co-infections (including hepatitis B virus (HBV), hepatitis C virus (HCV), tuberculosis, and sexually transmitted infections). Testing for HCV and HBV was carried out on blood samples, using the Anti-HCV-ELISA Best Kit (Vector-Best, Novosibirsk, Russia) and the HBsAg-ELISA Best Kit (Vector-Best, Novosibirsk, Russia), respectively. Patients reporting symptoms consistent with tuberculosis were screened for tuberculosis using chest X-rays.

### 2.2. Epidemiological Data

Epidemiological data were available for all participants, including the sample collection year, the sex of the participant, the region of residence, the reported transmission route, and ART regimens. To facilitate analysis, the 15 regions of residence were regrouped into 7 broader geographical regions (Figure 1, termed “Regions”), as follows: “Northern Kazakhstan”, which includes North Kazakhstan and Kostanay; “Southern Kazakhstan”, which includes Turkistan, Kyzylorda, and Zhambyl; “Eastern Kazakhstan”, which includes Pavlodar and East Kazakhstan; “Western Kazakhstan”, which includes West Kazakhstan, Atyrau, Mangystau, and Aktobe; “Central Kazakhstan”, which includes Karagandy; “Astana”, which includes the capital city Astana and Akmola, Astana’s surrounding region; and “Almaty”, which includes the city Almaty and its homonymous surrounding region.

Individual ART regimens were summarized as 1st-, 2nd-, or mixed-generation regimens according to the specific nucleoside reverse transcriptase inhibitors (NRTIs) and non-nucleoside reverse transcriptase inhibitors (NNRTIs) administered: zidovudine (AZT), didanosine (DDI), stavudine (D4T), and lamivudine (3TC) were classified as 1st-generation NRTIs; abacavir (ABC), tenofovir (TDF), and emtricitabine (FTC) were classified as 2nd-generation NRTIs; nevirapine (NVP) was classified as a 1st-generation NNRTI; and efavirenz (EFV), etravirine (ETR), and rilpivirine (RPV) were classified as 2nd-generation NNRTIs. Any protease inhibitors (PIs, darunavir/ritonavir, DRV/r; DRV/cobicistat, DRV/c; lopinavir/r, LPV/r) or integrase strand transfer inhibitors (INSTIs, dolutegravir, DTG) administered were not considered for the 1st-, 2nd-, or mixed-generation regimen classification, as subsequent analysis of DRMs only considered mutations in reverse transcriptase.

### 2.3. Sequence Generation

The mRNA from blood samples was purified using the Ribo-zol-C Kit (InterLabService, Moscow, Russia) and used for RT-PCR, followed by *pol* gene sequencing using AmpliSens^®^ HIV-Resist-Seq (Amplisens, Moscow, Russia) and Applied Biosystems^®^ Genetic Analyzer 3130 (Applied Biosystems Inc., Foster, RI, USA). Specifically, the sequence data consisted of partial HIV-1 *pol* gene sequences (covering protease and the 5′-end of reverse transcriptase, positions 2253 to 3337 on the HXB2 reference genome). After the removal of duplicates and sequence alignment, the final dataset consisted of 968 HIV-1 sequences from PLHIV in Kazakhstan.

### 2.4. Sequence Subtyping

HIV-1 sequences were subtyped using REGA v3 [18], with inconclusive sequences cross-checked against SCUEAL [19] and their location on a maximum likelihood phylogeny (generated using IQ-TREE v2.1.2 [20]).

### 2.5. Drug Resistance Profiling

The 968 HIV-1 *pol* sequences were screened for DRMs using the Stanford HIV Drug Resistance Database [21]. Specifically, the sequences were screened for mutations in protease and reverse transcriptase, and drug resistance scores (susceptible, potential low-, low-, intermediate-, and high-level resistance) were outputted by the Stanford HIVdb Program [22] for each of the sequences to the following 13/15 ARVs administered to patients in our dataset: 3TC, ABC, AZT, D4T, DDI, FTC, TDF, ETR, EFV, NVP, RPV, DRV/r, and LPV/r. Due to a lack of integrase sequence coverage, drug resistance profiling for DTG was not possible, while DRV/c is not an ARV screened for on the current version of the Stanford HIV Drug Resistance Database.

### 2.6. Phylogenetic Analysis

Phylogenetic trees were produced for subtypes A6 and CRF02_AG using Nextstrain [23]. Prior to phylogenetic analysis, 7 outgroup sequences for tree rooting were downloaded from the Los Alamos National Laboratories HIV Sequence Database [24] for each subtype, and sites of major DRMs (23 codon positions) were masked. Within the Nextstrain pipeline, maximum likelihood phylogenies were constructed using IQ-TREE [25], with substitution models automatically assigned using ModelFinder [26] (TVM + F + R7 for A6 and TVM + F + R6 for CRF02_AG). Time-resolved phylogenies were generated using TreeTime [27] with the least-squares method for rooting, and ancestral state reconstruction was performed for geographic regions and A62V transmission. The final trimmed A6 (*n* = 552) and CRF02_AG (*n* = 392) phylogenies were plotted in R v4.2.2 [28], using packages *ggtree* [29] and *ggtreeExtra* [30].

### 2.7. Statistical Analysis

All statistical analyses were carried out in R v4.2.2 [28], including a logistic stepwise regression model to determine which factors were associated with the presence of high-level drug resistance (with adjusted odds ratios, aORs, calculated for statistically significant variables among the crude ORs), and χ^2^ tests for heterogeneity (or Fisher’s exact test for smaller count values) to determine whether differences in observed variable frequencies were statistically significant.

## 3. Results

### 3.1. Demographic Characterization of Study Population

Between 2017 and 2020, 968 HIV-1 sequences were collected from PLHIV in Kazakhstan, with the highest number (389, 40.2%) collected in 2019 (Table 2). Of the sequenced participants, 468 (48.3%) were female, and participants’ ages ranged from 4 to 77 years (median = 41). The median time since HIV diagnosis was 8 years (range 1–22 years), the median time of ART was 4 years (range 0–15 years, with 7 individuals not undergoing ART), and the median time between diagnosis and ART initiation was 2 years (range 0–17 years, with only 268 (28.2%) individuals initiating ART within a year of diagnosis). Although according to *current* ART guidelines, only 12 (1.2%) participants were on preferred regimens and 471 (48.7%) were on alternative regimens, according to 2017–2020 guidelines, by the time that most participants were registered, 386 (39.9%) were on preferred regimens and 258 (26.7%) were on alternative regimens (Table 1 and Table 2). Almost half of participants were on second-generation ART regimens (464, 47.9%), followed by mixed- (291, 30.1%) and first-generation regimens (204, 21.1%). The combination of FTC + TDF + EFV was the most common (386, 39.9%) ART regimen administered (Table 2). Most participants reported no travel abroad (916, 94.6%) and were negative for reported or evident sexually transmitted infections (899, 92.9%). All participants were tested for other coinfections, with 409 (42.3%) individuals positive for HCV antibodies, 35 (3.6%) positive for HBV antigens, and 219 (22.6%) meeting Kazakh tuberculosis diagnostic criteria.

HIV-1 subtypes A6 and CRF02_AG made up almost the entire epidemic between them (954, 98.6%), with subtype A6 being the most prevalent in our dataset (553, 57.1%, Table 2). While subtype A6 and CRF02_AG sequences were sampled from all regions, the distribution of sequences across the regions differed significantly (*p*-value = 5 × 10^−4^), with CRF02_AG found predominantly in Almaty (175, 43.6%) and Southern Kazakhstan (86, 21.4%) and subtype A6 found more evenly across regions, with its highest frequency in Central Kazakhstan (148, 26.8%). Heterosexual HIV-1 infection was the most commonly reported transmission route (566, 58.5%), followed by transmission among PWID in 364 (37.6%) individuals (Table 2). The reported transmission routes also differed between subtypes A6 and CRF02_AG (*p*-value = 2.8 × 10^−5^), with heterosexual transmission more commonly reported alongside subtype A6 infections (63.1% vs. 52.1%) and PWID transmission more commonly reported alongside CRF02_AG infections (44.1% vs. 33.3%). When comparing heterosexual and PWID transmission routes, men were more likely to acquire HIV through injecting drug use (60.2%), while women were more likely to do so through heterosexual contacts (83.3%, *p*-value < 2.2 × 10^−16^). For more recent diagnoses, the probability of acquiring HIV through PWID transmission routes decreased by 11% compared to heterosexual routes for every year change in diagnosis time (*p*-value = 5.88 × 10^−14^).

Amongst reported heterosexual transmissions (*n* = 566), risk factors were assessed for 472 (83.4%) participants. Of these, no risk group was registered for 326 (69.1%, 33.7% of all 968 study participants) individuals, while 3 (0.6%, 0.3% of total) participants were recorded as sex workers and 143 (30.3%, 14.8% of total) were recorded as having had sexual contact with either sex workers, PWID, or PLHIV.

### 3.2. Drug Resistance Profiles

Over one-third (355, 36.7%) of the HIV-1 study sequences exhibited low-to-high-level drug resistance to at least one of the 13 ARVs screened for, with almost all exhibiting high levels of drug resistance (311, 87%). Of those with drug resistance, 304/355 (85.6%) participants had sequences resistant to at least one drug on their ART regimen, while 282/311 (90.7%) participants had sequences with high-level resistance to at least one drug on their ART regimen. High-level NNRTI resistance was most prevalent (291, 30.1%), followed by high-level NRTI resistance (163, 16.8%). Only 2 sequences had high-level PI resistance, and both high-level NRTI and NNRTI resistance mutations were exhibited in 143 (14.8%) sequences (Table 2).

To determine whether certain factors were associated with the presence of high-level drug resistance, a logistic stepwise regression model was used. The best-fitting model (AIC: 1170.4) included ART generation, CD4+ T-cell count, and the time undergoing ART; the odds of high-level drug resistance were higher for individuals on second- (aOR: 2.74, 95% CI: 1.83–4.16) and mixed-generation (aOR: 1.53, 95% CI: 1.00–2.37) regimens compared to those on first-generation regimens. No significant differences in the frequency of high-level drug resistance were found between subtypes A6 and CRF02_AG; however, some differences in individual ARV use were detected, including a moderately higher use of NVP in patients with CRF02_AG infections (Appendix A). Individual ARV use significantly differed in 7/15 ARVs between patients with or without high-level drug resistance (Appendix A), with FTC, TDF, and EFV (all second-generation ARVs) having been more frequently taken by patients with high-level drug resistance.

### 3.3. Distinct DRMs among Subtypes A6 and CRF02_AG

Our 968-sequence dataset included 364 (37.6%) sequences with NRTI-associated DRMs, 340 (35.1%) with NNRTI-associated DRMs, and 12 (1.2%) with major PI-associated DRMs. Of these, 101/364 (27.7%) had ≥two NRTI-associated DRMs (excluding A62V) and 152/340 (44.7%) had ≥two NNRTI-associated DRMs. Due to the high proportion of sequences with ≥ two NRTI/NNRTI-associated DRMs, we tested whether the frequency of certain double DRMs was higher than expected when assuming each mutation arises independently. We tested this for an NRTI pair of DRMs (M184V and K65R) and an NNRTI pair (K103N and G190S); for M184V and K65R, the frequency of double DRMs was significantly higher than expected (4%, *p*-value = 0.015), while for K103N and G190S, the frequency was significantly lower (0.9%, *p*-value = 6.2 × 10^−4^).

In total, 22 DRMs were present at over 1% frequency, which are summarized according to the subtype in Table 3 and according to ART generation in Appendix A. Of these, eight DRMs were present at over 4% frequency, plotted according to subtype and ART generation in Figure 2A,B, respectively. The frequencies of certain DRMs were significantly different between subtypes A6 and CRF02_AG; accessory NRTI-associated mutation A62V was more frequently observed in subtype A6, alongside major NRTI mutation D67N, major NNRTI mutation G190S, and accessory NNRTI mutation K101E, while major NNRTI-associated mutation K103N and accessory NNRTI mutation V179E were more frequently observed in CRF02_AG (Table 3, Figure 2A). Of the eight DRMs present at over 4% frequency, five were significantly more common amongst individuals on second-generation ART regimens, corroborating the findings of more frequent high-level drug resistance amongst second-generation regimens (Figure 2B); however, individual ARVs did not significantly affect the presence of any of the 22 DRMs.

For the six DRMs with significantly different frequencies between subtypes, we investigated differences in codon usage between pairs to determine if codon usage differences explained the differing DRM frequencies (Table 4). For K103N and K101E, no significant differences in codon usage were found between subtypes A6 and CRF02 in sequences without the respective DRM. For A62V and D67N, codon usage frequencies significantly differed among the sequences without the DRM; however, for each site, only one nucleotide change was required in each codon for DRM presence. For G190S and V179E, codon usage frequencies significantly differed among the sequences without the DRM, with codons requiring two nucleotide changes for DRM acquisition (as opposed to a single one) being more common in the subtype with a lower DRM frequency (G190S for CRF02_AG, V179E for A6). This suggests that different mutational pathways may affect the significantly different frequencies of G190S and V179E in subtypes A6 and CRF02_AG.

### 3.4. Phylogenetic Analysis of Subtypes A6 and CRF02_AG

Time-resolved phylogenies for subtypes A6 (Figure 3A) and CRF02_AG (Figure 3B) were produced using Nextstrain. The time to the most recent common ancestor (tMRCA) for the subtype A6 sequences was 10 years earlier than for the CRF02_AG ones (tMRCA = 1983 vs. tMRCA = 1993, respectively). The A6 phylogeny showed a deep split in the tree early in the epidemic (1989) from where 98.7% of sequences originated, indicating an early separation between sequences originating within Central Kazakhstan and spreading to Western Kazakhstan (Figure 3A, bottom) vs. sequences spreading through Southern and Eastern Kazakhstan (Figure 3A, top), with the latter sequences overwhelmingly harboring DRM A62V. No single major split was observed in the CRF02_AG phylogeny. Instead, ancestral state reconstruction (not shown) indicated the presence of multiple lineages descended from a common ancestor in Almaty, emerging in other regions at later stages in the epidemic. Other than the clear clustering of A62V mutations in the A6 phylogeny, no other distinct patterns of DRM transmission were observed. For CRF02_AG (Figure 3B) and, to a lesser extent, subtype A6 (Figure 3A), the presence of high-resistance sequences on the phylogeny mirrors the presence of K103N, a major NNRTI-associated DRM.

## 4. Discussion

In this study, we analyzed 968 HIV-1 *pol* sequences from ART-receiving individuals from all regions of Kazakhstan, covering almost 3% of the estimated number of PLHIV in 2020. Our aim was to quantify levels of HIV-1 drug resistance, identify DRMs, and elucidate the evolutionary history of the HIV-1 epidemic in Kazakhstan using a phylogenetic approach (including the appearance and spread of DRMs within Kazakhstan). Among sequenced participants, the Kazakh HIV-1 epidemic is almost exclusively comprised (954, 98.6%) of two subtypes, A6 and CRF02_AG, with the frequencies of each one varying across geographical regions in the country. Almost one-third (311, 32.1%) of sequences exhibited high-level drug resistance; however, distinct DRMs were observed among subtypes A6 and CRF02_AG. Phylogenetic analysis revealed differences between the two subtypes regarding their introduction into Kazakhstan, both in timing and location. Our findings provide insights into the molecular epidemiology of HIV-1 in Kazakhstan, a country in Central Asia where HIV incidence continues to increase. 

### 4.1. Demographics of the HIV-1 Epidemic

While samples were collected from all regions of Kazakhstan, sequences from Almaty, Central, and Southern Kazakhstan were overrepresented compared to other regions (Table 2), as they constitute the more populous regions of the country [31]. In the WHO region of Eastern Europe and Central Asia, 91% of new HIV infections as of 2020 were among key populations, including PWID (43%), sex workers (13%), MSM (16%), and sex partners of all key populations (18%) [3]. In our study, PWID transmission was reported in 37.6% of participants, lower than WHO regional estimates but higher than the 30.1% prevalence reported for surveys conducted between 2002 and 2013, where Kazakhstan was identified as the country with the highest HIV prevalence in Central Asia and the Caucasus [32]. In total, 14.8% of individuals reported heterosexual transmission associated with sexual contact with key populations; however, sex workers (0.3%) and MSM (0.5%) were underrepresented in our dataset, likely due to the stigma associated with reporting. Indeed, 33.7% of heterosexual transmissions in our study had no associated risk factor recorded, which could be a sign of underreporting of risk behaviors and/or of a shifting epidemic, towards the general population. According to official statistics, 66% of HIV infections in Kazakhstan in 2008 were among PWID, whereas the current transmission trends in 2021 are increasingly shifting toward heterosexual and MSM populations, constituting, respectively, 65.7% and 13.6% of all PLHIV [33]. This may reflect successes in risk reduction for PWID in the country [34,35]. Our results indicate that men are more likely to acquire HIV through PWID transmission, while women are more likely to do so through heterosexual contact, consistent with findings from the HIV epidemic in Kyrgyzstan [17]. Among our participants, the number of patients infected with subtype A6 was 1.4 times higher than CRF02_AG (Table 2). Historically, subtype A6 is known to have originated in Central Asia and former Soviet Union (FSU) countries [16], and still remains the most predominant HIV variant in this region, including Kazakhstan, through autochthonous transmission [36,37].

### 4.2. High-Level Drug Resistance and Distinct DRMs between the Subtypes

Our study reports high frequencies (32.1%) of high-level drug resistance among PLHIV in Kazakhstan. Prior studies had smaller sample sizes. A study of 85 isolates collected between 2001 and 2003 across Kazakhstan found no primary DRMs associated with the sequences [14]. Another study of 165 ART-naïve patients, with sequences collected between 2009 and 2013, reported a low (3%) prevalence of drug resistance [13]. A study of 602 sequences collected between 2017 and 2019 from ART-experienced patients reported high-level NRTI resistance in 10% of sequences and high-level NNRTI resistance in 21% of them [7]. Comparing these earlier prevalence estimates to our findings of 16.8% and 30% high-level NRTI and NNRTI resistance, respectively, our current study suggests an increase in the prevalence of drug resistance in Kazakhstan over time. Using our logistic regression model to determine whether certain factors were associated with the presence of high-level drug resistance, we found the odds of resistance to be higher for individuals on ART regimens containing second-generation ARVs. While it is possible that the administration of second-generation ARVs is causing high levels of drug resistance in Kazakhstan, it is even more likely that the observed correlation is a reflection of patients being prescribed second-generation ARVs after developing high-level resistance and thus failing therapy with first-generation ART regimens.

The prevalence of high-level drug resistance in our study did not significantly differ between subtypes A6 and CRF02_AG. However, distinct DRM frequencies were observed between the two subtypes, namely a higher prevalence of A62V, G190S, K101E, and D67N in subtype A6, and a higher prevalence of K103N and V179E in CRF02_AG (Table 3). Despite the comparatively higher prevalence of K103N in CRF02_AG, the mutation was still the most prevalent major NNRTI-associated DRM in subtype A6 as well. The high frequency of this mutation within our study population is of concern due to the high-level NNRTI resistance that this causes, reducing NVP and EFV susceptibility by about 50- and 20-fold, respectively [21,38]. A study conducted in Russia, where subtype A6 predominates, revealed that A62V was prevalent in 63% of ART-naïve HIV cases [39]; a second Russian study reported that DRMs A62V and G190S were significantly associated with subtype A6, similar to our findings [40]. To determine if codon usage differences explained the differing DRM frequencies between the two subtypes, we carried out a codon usage analysis (Table 4). This suggested that different mutational pathways may be affecting the appearance of G190S and V179E. A previous study from Russia corroborates our findings for G190S: the mutation was not found in CRF02_AG compared to subtype A6 (*p*-value < 0.001) and the DRM in A6 resulted from a single nucleotide transition (G to A), a concerning predisposition due to the high level of NNRTI resistance caused by the mutation [41].

### 4.3. Evolutionary History of the HIV-1 Epidemic in Kazakhstan

Phylogenetic analyses in Nextstrain revealed that the tMRCA for subtype A6 sequences was earlier than for CRF02_AG sequences (1983 vs. 1993, respectively, Figure 3), with a deep split in the tree in 1989, from where 98.7% of the sequences originate and spread to distinct regions. This split is consistent with the chronology of HIV-1 subtype A6 in Kazakhstan; HIV-1 transmission spread through FSU countries after the downfall of the Soviet Union in 1991, with the first documented HIV outbreak in Kazakhstan in 1996 in Temirtau, a city in Central Kazakhstan [42,43,44]. There are no prior studies investigating the origins of CRF02_AG in Kazakhstan. Our analysis revealed that the oldest CRF02_AG infections may have occurred in Almaty, from where they spread to other regions. This, alongside the overwhelming frequency of CRF02_AG in Almaty and Southern Kazakhstan, is consistent with findings characterizing the HIV-1 epidemic in Kyrgyzstan, directly south of Kazakhstan, where CRF02_AG dominates the epidemic [17].

### 4.4. Limitations

Along with strengths, our study has some limitations. Firstly, most sequences analyzed in this study were obtained from ART-experienced PLHIV and we were unable to (a) estimate the prevalence of drug resistance in ART-naïve PLHIV and (b) distinguish between acquired or transmitted DRMs though phylogenies that provide evidence for A62V transmission in subtype A6. While tools such as Nextstrain provide rapid estimates for phylogenetic analysis of epidemics, the results may be less accurate than those of more computationally intensive programs.

## 5. Conclusions

Our study describes the molecular epidemiology of the HIV-1 epidemic in Kazakhstan, characterized by high levels of drug resistance and distinct DRMs between subtypes A6 and CRF02_AG, with some (G190S and V179E) suggesting that different mutational pathways may be contributing to the distinct rates of DRM acquisition between the two variants. The high levels of resistance exemplify the need for drug resistance monitoring within Kazakhstan, including DRM screening at diagnosis, while distinct DRMs between subtypes argue for the inclusion of subtyping within standard diagnostic protocols. These practices will inform the prescription of the most effective ART regimens, thus helping to prevent the emergence of new DRMs.

## Figures and Tables

**Figure 1 viruses-15-01407-f001:**
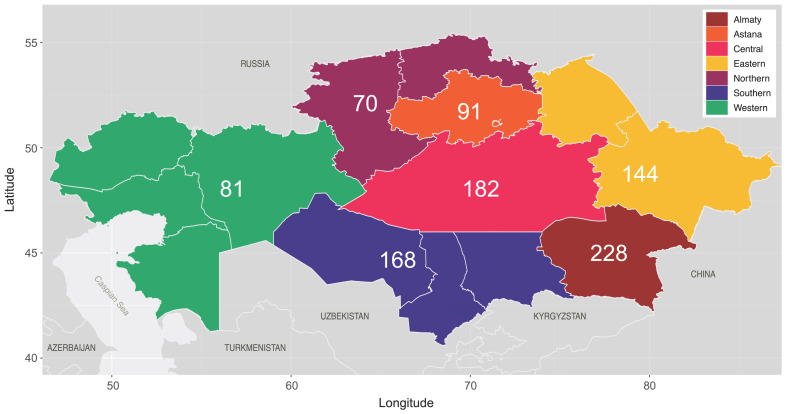
Regional map of Kazakhstan, according to 2020 administrative divisions. Regions are grouped and colored according to broader geographical regions of sample collection, with the numbers denoting the number of sequences sampled from each grouped region (*n* = 964, NA = 4).

**Figure 2 viruses-15-01407-f002:**
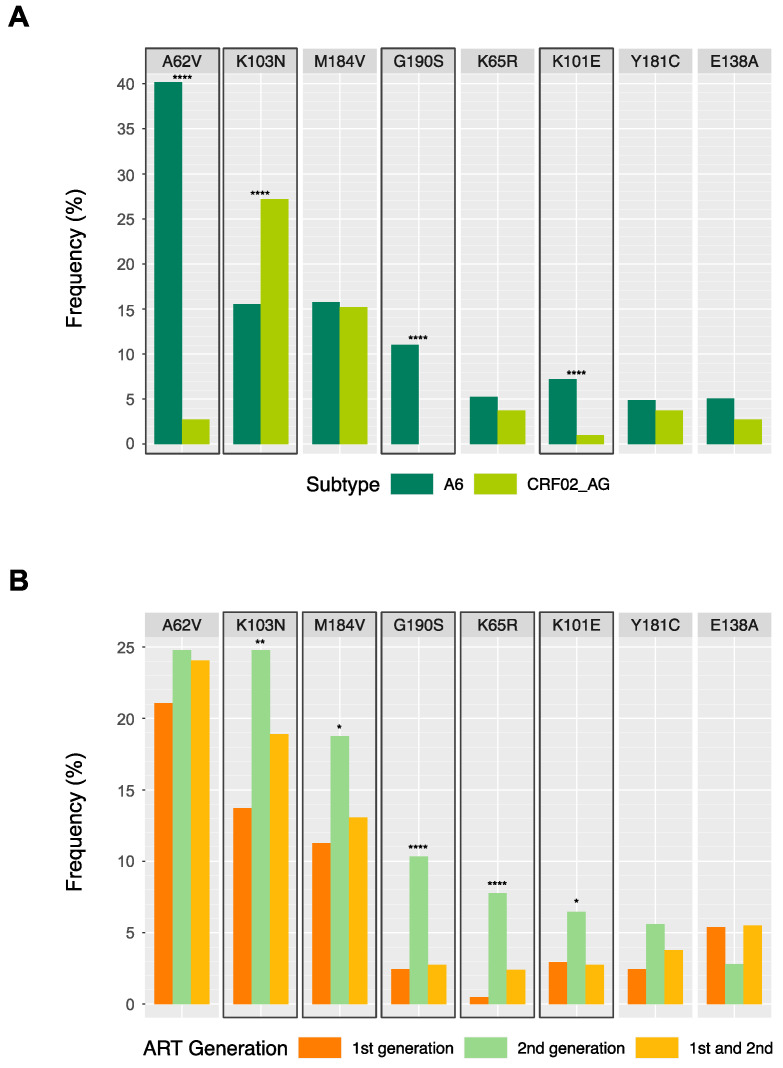
Drug-resistant mutations (DRMs) present at over 4% frequency among individuals in the Kazakhstan dataset (*n* = 968), according to HIV subtype (**A**) and antiretroviral therapy (ART) generation (**B**). Highlighted boxes indicate significant differences between categories based on χ^2^ tests for heterogeneity, with asterisks indicating *p*-value cut-offs (* = *p*-value < 0.05; ** = *p*-value < 0.01; **** = *p*-value < 0.0001).

**Figure 3 viruses-15-01407-f003:**
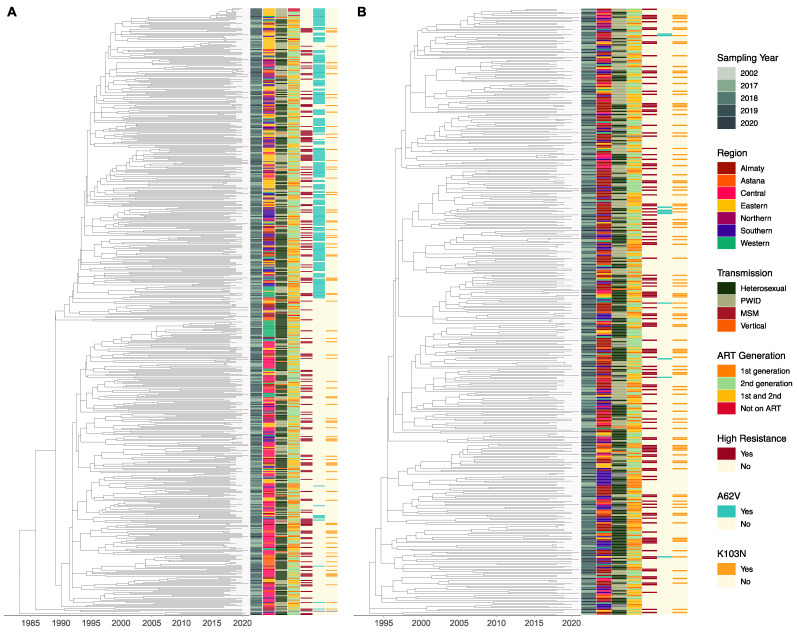
Time-resolved subtype A6 (**A**, *n* =552, tMRCA = 1983) and circulating recombinant from CRF02_AG (**B**, *n* = 392, tMRCA = 1993) trees. Stacked tiles are colored from left to right according to sequence sampling year, the region of sample collection, reported transmission route, ART generation, high-level drug resistance, and the presence of drug-resistant mutations A62V and K103N, with any missing data colored in grey. Trees were trimmed to the start of Kazakh epidemic for each subtype, removing any outliers and reference genomes used for rooting. PWID = people who inject drugs; MSM = men who have sex with men; ART = antiretroviral therapy.

**Table 1 viruses-15-01407-t001:** Guidelines for preferred and alternative antiretroviral therapy regimens in Kazakhstan.

	*Preferred*	*Alternative*	*Sources*
*2017–2020*	3TC/FTC + TDF + EFV	3TC + AZT + EFV/NVP	[5,6]
		3TC/FTC + TDF + DTG/NVP	
*2020–present*	3TC/FTC + TDF/TAF + DTG/BIC	3TC/FTC + TDF + EFV	[7,8]
	3TC + DTG	3TC + ABC + EFV/DTG	
		3TC/FTC + ABC/TDF + RAL	
		3TC/FTC + TDF/TAF + EVG/c	
		3TC/FTC + TDF/TAF/ABC + DRV/c/DRV/r	

3TC: lamivudine, ABC: abacavir, AZT: zidovudine, BIC: bictegravir, DRV/c: darunavir/cobicistat, DRV/r: darunavir/ritonavir, DTG: dolutegravir, EFV: efavirenz, EVG/c: elvitegravir/cobicistat, FTC: emtricitabine, NVP: nevirapine, RAL: raltegravir, TAF: tenofovir alafenamide, TDF: tenofovir.

**Table 2 viruses-15-01407-t002:** Demographic distribution of HIV-1 sequences in Kazakhstan, according to subtype (*n* = 968).

	*A6*	*CRF02_AG*	*Other*	*Total*
	*n (%)*	*n (%)*	*n (%)*	*n (%)*
*All sequences*	553 (57.1)	401 (41.4)	14 (1.4)	968 (100)
*Gender*				
Female	272 (49.2)	190 (47.4)	6 (42.9)	468 (48.3)
Male	281 (50.8)	211 (52.6)	8 (57.1)	500 (51.7)
*Sampling year*				
2017	124 (22.4)	104 (25.9)	1 (7.1)	229 (23.7)
2018	138 (25)	110 (27.4)	1 (7.1)	249 (25.7)
2019	240 (43.4)	140 (34.9)	9 (64.3)	389 (40.2)
2020	51 (9.2)	47 (11.7)	3 (21.4)	101 (10.4)
*Region*				
Almaty	49 (8.9)	175 (43.6)	4 (28.6)	228 (23.6)
Astana	58 (10.5)	33 (8.2)	0	91 (9.4)
Central Kazakhstan	148 (26.8)	33 (8.2)	1 (7.1)	182 (18.8)
Eastern Kazakhstan	116 (21)	26 (6.5)	2 (14.3)	144 (14.9)
Northern Kazakhstan	45 (8.1)	21 (5.2)	4 (28.6)	70 (7.2)
Southern Kazakhstan	80 (14.5)	86 (21.4)	2 (14.3)	168 (17.4)
Western Kazakhstan	56 (10.1)	24 (6)	1 (7.1)	81 (8.4)
NA	1 (0.2)	3 (0.7)	0	4 (0.4)
*Transmission* ^1^				
Heterosexual	349 (63.1)	209 (52.1)	8 (57.1)	566 (58.5)
PWID	184 (33.3)	177 (44.1)	3 (21.4)	364 (37.6)
MSM	3 (0.5)	0	2 (14.3)	5 (0.5)
Vertical	13 (2.4)	3 (0.7)	1 (7.1)	17 (1.8)
NA	4 (0.7)	12 (3)	0	16 (1.7)
*ART regimen* ^2^				
FTC + TDF + EFV	231 (41.8)	148 (36.9)	7 (50)	386 (39.9)
3TC + AZT + EFV	58 (10.5)	48 (12)	0	106 (11)
3TC + AZT + NVP	60 (10.8)	65 (16.2)	0	125 (12.9)
Other	196 (35.4)	139 (34.7)	7 (50)	342 (35.3)
Not on ART	7 (1.3)	0	0	7 (0.7)
NA	1 (0.2)	1 (0.2)	0	2 (0.2)
*High-level resistance* ^3^				
Any	172 (31.1)	135 (33.7)	4 (28.6)	311 (32.1)
NRTI only ^4^	11 (2)	9 (2.2)	0	20 (2.1)
NNRTI only	72 (13)	73 (18.2)	3 (21.4)	148 (15.3)
NRTI + NNRTI ^5^	89 (16.1)	53 (13.2)	1 (7.1)	143 (14.8)

^1^ PWID: persons who inject drugs, MSM: men who have sex with men. ^2^ ART: antiretroviral therapy, 3TC: lamivudine, AZT: zidovudine, EFV: efavirenz, FTC: emtricitabine, NVP: nevirapine, TDF: tenofovir. ^3^ NRTI: nucleoside reverse transcriptase inhibitor, NNRTI: non-nucleoside reverse transcriptase inhibitor. ^4^ Includes one A6 sequence with NRTI and protease inhibitor (PI) resistance. ^5^ Includes one CRF02_AG sequence with NRTI + NNRTI + PI resistance.

**Table 3 viruses-15-01407-t003:** Drug-resistant mutations (DRMs) present at over 1% frequency, according to subtype (*n* = 968).

	*A6*	*CRF02_AG*	*Other*	*Total*
	*n (%)*	*n (%)*	*n (%)*	*n (%)*
*All sequences*	553 (57.1)	401 (41.4)	14 (1.4)	968 (100)
*NRTI major*				
M184V	87 (15.7)	61 (15.2)	1 (7.1)	149 (15.4)
K65R	29 (5.2)	15 (3.7)	1 (7.1)	45 (4.6)
**D67N**	16 (2.9)	1 (0.2)	0	17 (1.8)
K70R	10 (1.8)	5 (1.2)	1 (7.1)	16 (1.7)
Y115F	9 (1.6)	6 (1.5)	0	15 (1.5)
M41L	8 (1.4)	6 (1.5)	0	14 (1.4)
K70E	9 (1.6)	4 (1)	0	13 (1.3)
L74I	10 (1.8)	3 (0.7)	0	13 (1.3)
L74V	7 (1.3)	4 (1)	0	11 (1.1)
*NRTI accessory*				
**A62V**	222 (40.1)	11 (2.7)	0	233 (24.1)
K219E	13 (2.4)	10 (2.5)	0	23 (2.4)
*NNRTI major*				
**K103N**	86 (15.6)	109 (27.2)	4 (28.6)	199 (20.6)
**G190S**	61 (11)	0	0	61 (6.3)
Y181C	27 (4.9)	15 (3.7)	0	42 (4.3)
P225H	15 (2.7)	15 (3.7)	1 (7.1)	31 (3.2)
G190A	5 (0.9)	8 (2)	0	13 (1.3)
K238T	3 (0.5)	7 (1.7)	0	10 (1)
*NNRTI accessory*				
**K101E**	40 (7.2)	4 (1)	0	44 (4.5)
E138A	28 (5.1)	11 (2.7)	1 (7.1)	40 (4.1)
A98G	11 (2)	5 (1.2)	0	16 (1.7)
**V179E**	3 (0.5)	11 (2.7)	1 (7.1)	15 (1.5)
V108I	10 (1.8)	4 (1)	0	14 (1.4)

NRTI: nucleoside reverse transcriptase inhibitor, NNRTI: non-nucleoside reverse transcriptase inhibitor. DRMs in **bold** denote statistically significant differences between their frequencies in subtype A6 and CRF02_AG, specifically: D67N (*p*-value = 0.005), A62V (*p*-value < 2 × 10^−16^), K103N (*p*-value = 1.6 × 10^−5^), G190S (*p*-value = 1.6 × 10^−11^), K101E (*p*-value = 1.2 × 10^−5^), and V179E (*p*-value = 0.01).

**Table 4 viruses-15-01407-t004:** Codon usage for sites where subtypes A6 (*n* = 553) and CRF02_AG (*n* = 401) significantly differ in drug-resistant mutation (DRM) frequencies.

			*A6*	*CRF02_AG*
			*n (%)*	*n (%)*
** *A62V* **	*DRM* present	GTT	191 (86)	0
		GTC	9 (4.1)	8 (72.7)
		Other	22 (9.9)	3 (27.3)
	*DRM absent* (χ^2^ test *p*-value < 2 × 10^−16^)	GCT	318 (96.1)	22 (5.6)
		GCC	4 (1.2)	347 (89)
		Other	9 (2.7)	21 (5.4)
** *K103N* **	*DRM present*	AAC	54 (62.8)	65 (59.6)
		Other	32 (37.2)	44 (40.4)
	*DRM absent*	AAA	438 (93.8)	275 (94.2)
		Other	29 (6.2)	17 (5.8)
** *G190S* **	*DRM present*	AGC	53 (86.9)	0
		Other	8 (13.1)	0
	*DRM absent* (χ^2^ test *p*-value < 2 × 10^−16^)	GGC	449 (91.3)	10 (2.5)
		GGG	3 (0.6)	358 (89.3)
		Other	40 (8.1)	33 (8.2)
** *K101E* **	*DRM present*	GAA	36 (90)	2 (50)
		Other	4 (10)	2 (50)
	*DRM absent*	AAA	472 (92)	369 (92.9)
		Other	41 (8)	28 (7.1)
** *D67N* **	*DRM present*	AAT	7 (43.8)	1 (100)
		AAC	6 (37.5)	0
		Other	3 (18.8)	0
	*DRM absent* (χ^2^ test *p*-value < 2 × 10^−16^)	GAT	265 (49.3)	367 (91.8)
		GAC	257 (47.9)	17 (4.3)
		Other	15 (2.8)	16 (4)
** *V179E* **	*DRM present*	GAA	1 (33.3)	9 (81.8)
		GAG	1 (33.3)	0
		Other	1 (33.3)	2 (18.2)
	*DRM absent* (χ^2^ test *p*-value < 2 × 10^−16^)	GTT	413 (75.1)	4 (1)
		GTA	2 (0.4)	295 (75.6)
		Other	135 (24.5)	91 (23.3)

*Note:* codons with ambiguous nucleotides are all classified as “Other”; χ^2^ test *p*-values are only provided for significant differences in codon usage where the specified DRM is absent.

## Data Availability

Study data are unavailable to protect patient confidentiality.

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
