# Peer review of "High-Level Drug-Resistant Mutations among HIV-1 Subtype A6 and CRF02_AG in Kazakhstan"

_viruses, 2023, doi:10.3390/v15071407_

Round 1
Reviewer 1 Report
Mukhatayeva et al., in their study entitled “High-level drug-resistant mutations among HIV-1 subtype A6 and CRF02_AG in Kazakhstan”, address the concerning 73% increase in HIV-1 infections in Kazakhstan between 2010 to 2020. Sequencing 968 patients country-wide, the authors find that subtype A6 accounts for 57% and CRF02_AG accounts for 41% of PLHIV. Drug resistance mutations for protease and pol were analyzed showing that the two subtypes had 1) high levels of drug-resistance, 2) exhibited different mutation profiles, and 3) showed different timing and geographical spread via phylogenetic analysis. A very nice study compiling the mutations and spread of HIV in Kazakhstan. The paper is well written, and the data support their findings indicating the need to genotype patients in Kazakhstan for effective HIV infection management.
Considerations:
1. For readers not familiar with Kazakhstan, it would help the reader to provide a map of the country with maybe “dotted lines?” indicating how you designated the regions (south, central, where Almaty is, etc). Could label in each region the number of patients analyzed.
2. Did you identify any recombinants between A6 and CRF02_AG? Low frequency would also support the different geographical spread of the two subtypes.
3. Bottom of page 3: For pol gene sequencing where you state, “1 kilobase, covering protease and the 5’ end of pol”… you should also provide the specific nucleotide position number sequenced versus HXB2 reference strain.
4. On page 4: For “drug resistance scores (susceptible, potential low-, low-, intermediate-, and high-level resistance)”… how were these designations defined? Segregated by the number of mutations? And/or by a specific mutation? Define for the reader.
5. Likewise, in Figure 2 the “high resistance column” closely mirrors the “K103N column”. Is K103N the only defining mutation for high resistance? Could provide references for K103N as a clinically important mutation leading to 20-50 fold resistance to most NNRTIs.
6. Figure 2: Legend is so small to read. I think it would benefit to enlarge legend/stretch lengthwise to be able to view the data better. It is such a nice example of tracking viral spread.
Author Response
Please see attached response letter.

Reviewer 2 Report
Comments for the manuscript by Mukhatayeva et al.:
In this study, the authors evaluated the mutations in Kazakhstan that are resistant to ART regimens. HIV-1 subtype A6 and CRF02_AG that are most prevalent in Kazakhstan showed various degrees of mutations in protease and reverse transcriptase that are expected to cause resistance to ART.
This study is very interesting and representative by using 968 patient samples. The use of first-generation, second-generation and mix ART regimens was also recorded. The study design and inclusion of patient samples are adequate. The data are generally easy to understand. Overall, this is an excellent study that reports mutation that would confer resistance to different ART regimens in different region of Kazakhstan.
There are some suggestions to improve the manuscript:
Statistics analyses are included for the tables and figures. I found one error that should be corrected: Both “***” and “****” are presented as “=p<0.001” in the legend to Figure 1.
The authors used “% frequency” in the text. However, “Proportion” is used for the y-axis in Figure 1, Figure 2 and Figure S1. Presenting the y-axis in the figure as “frequency (%)” would be consistent and clear to the readers.
Overall, this is an excellent study that will help to design more effective ART combinations to control HIV in Kazakhstan.
Author Response
Please see attached response letter.
